# A DIMENSIONAL ANALYSIS OF VIDEO ANOMALY DETECTION BENCHMARKS

## ABSTRACT

Benchmark datasets have fueled advances in video anomaly detection, yet they often embed hidden assumptions that distort both research focus and real-world applicability. Common benchmarks implicitly assume that anomalies are human-centric, visually salient, short-lived, and unambiguous to label, while neglecting object-driven, contextual, long-term, or ethically sensitive events. To expose and systematize these biases, we conduct the first dimensional analysis of anomaly detection benchmarks. Our framework organizes dataset design along four principled axes: content (*e.g.*, taxonomy, motion, modality), annotation (*e.g.*, density, human involvement, consistency), distribution (*e.g.*, frequency, diversity, temporal extent), and societal impact (*e.g.*, privacy, fairness). Applying this framework, we uncover structural imbalances: most benchmarks overrepresent conspicuous human anomalies while underrepresenting subtle or multimodal patterns, along with inconsistent annotation protocols and skewed anomaly distributions that confound fair evaluation. These design choices restrict the diversity of learnable patterns, bias algorithmic search spaces, and limit the operational robustness of deployed systems. We consolidate our findings into actionable guidelines for next-generation benchmarks that broaden anomaly coverage, enable reproducible evaluation, and embed social responsibility into dataset design. By reframing benchmarks through a dimensional lens, this work lays the foundation for more generalizable, equitable, and trustworthy video anomaly detection.

## 1 INTRODUCTION

Video anomaly detection (VAD) has emerged as a central challenge in video understanding, with critical applications spanning public safety, traffic monitoring, healthcare, retail, and transportation (Ding & Wang, 2025; Zhu et al., 2024; Samaila et al., 2024). Unlike canonical tasks such as classification or recognition, anomaly detection targets rare, diverse, and context-dependent events that deviate from normal patterns. This open-ended nature makes VAD uniquely difficult: anomalies are infrequent, weakly defined, and demand generalization far beyond the training distribution. In this landscape, datasets play a decisive role. By codifying what counts as an anomaly, they implicitly shape model objectives, steer algorithmic design, and influence the direction of the research field itself (Gebru et al., 2021). Yet, despite their pivotal role, existing VAD benchmarks embody hidden assumptions that constrain progress. Many popular datasets equate anomalies with conspicuous, human-centric events, *e.g.*, fighting or running (Sultani et al., 2018b), while overlooking object-driven, contextual, or multimodal anomalies that often arise in real-world environments (Wu et al., 2020; Thakare et al., 2023). Other datasets (Cheng et al., 2021; Ramachandra & Jones, 2020) assume anomalies are visually obvious, short-lived, and consistently labelable, neglecting subtler shifts that require contextual reasoning or long-term temporal modeling. These implicit choices embed structural biases: they narrow the operational scope of detection, limit the diversity of learnable patterns, and complicate fair comparison across methods (Ott et al., 2022; Zendel et al., 2017).

To address these challenges, we propose a dimensional framework for systematically analyzing and characterizing VAD benchmarks. Our framework organizes dataset design along four principled axes: *content* (*e.g.*, anomaly taxonomy, motion complexity, sensing modality), *annotation* (*e.g.*, density, human involvement, consistency), *distribution* (*e.g.*, frequency, diversity, temporal extent), and *societal impact* (*e.g.*, privacy, fairness). This perspective moves beyond ad hoc critiques to

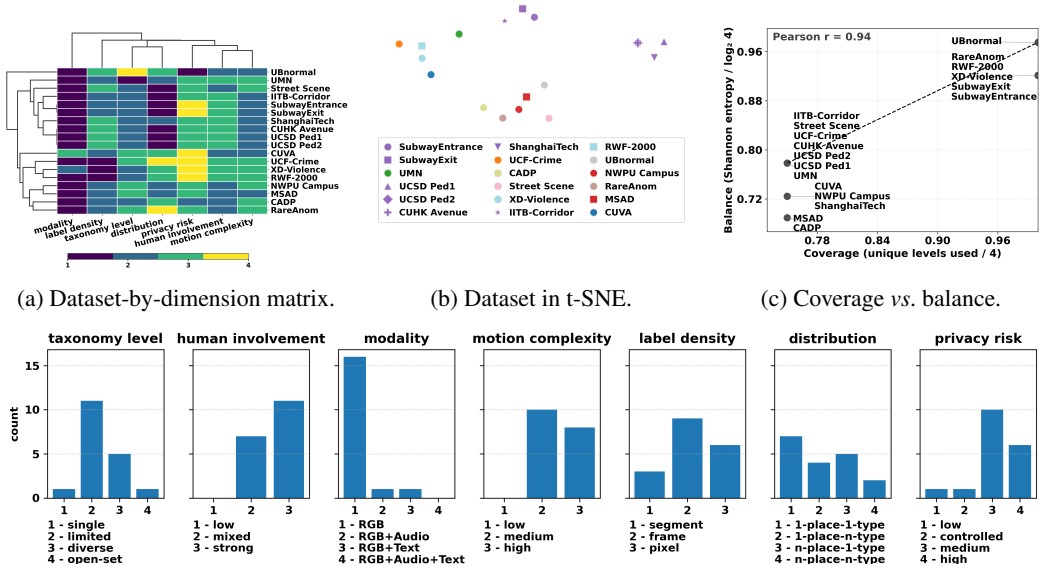

(a) Dataset-by-dimension matrix.  (b) Dataset in t-SNE.  (c) Coverage *vs*. balance.

(d) Marginal distributions of the seven design dimensions across datasets.

Figure 1: Dimensional overview of VAD benchmarks. (a) Clustered dataset-by-dimension matrix using Euclidean distance and average linkage (rows and columns), visualized as a discrete viridis heatmap (levels 1-4). Only fully rated rows were included; missing entries are excluded from clustering and counts. (b) Standardized 7D ordinal vectors (z-scored) embedded in 2D via t-SNE; each marker represents a dataset, colored by family. Pairwise distances reflect similarity across seven design axes: taxonomy, human involvement, modality, motion complexity, label density, distribution, and privacy risk. (c) Coverage is the fraction of distinct levels used across the seven dimensions (range $[0, 1]$); Balance is the normalized Shannon entropy of level usage (range $[0, 1]$). Pearson correlation between coverage and balance is reported; the dashed curve shows an upper envelope connecting maximum Balance per six equal-width coverage bins. (d) Taxonomy: single (one anomaly type), limited (few fixed types), diverse (many closed-set types), open-set (novel/unknown types); human involvement: low (minimal humans), mixed, strong (human-driven anomalies); modality: RGB, RGB+Audio, RGB+Text, RGB+Audio+Text; motion complexity: low (static/predictable), medium, high (dynamic, multi-agent); label density: segment, frame, pixel; distribution: 1-place-1-type, 1-place-n-type, n-place-1-type, n-place-n-type; privacy risk: low, controlled, medium, high.

provide a structured lens for identifying biases, uncovering blind spots, and guiding the creation of more representative benchmarks. This paper makes three main **contributions**:

i. We introduce the first *dimensional framework* for analyzing video anomaly detection benchmarks, capturing how implicit design choices shape the scope and difficulty of the task.

ii. We apply this framework to 18 widely used datasets, including long-standing benchmarks such as UCF-Crime (Sultani et al., 2018b) and ShanghaiTech (Luo et al., 2017), as well as recent datasets like MSAD (Zhu et al., 2024) and CUVA (Lu et al., 2013), revealing systematic biases and underexplored regions of the design space.

iii. We demonstrate how these structural imbalances affect model evaluation and generalization, and we propose actionable principles for constructing future benchmarks that broaden anomaly coverage, enable reproducible evaluation, and align more closely with real-world deployment demands.

Through this dimensional analysis, we not only consolidate the fragmented landscape of existing benchmarks but also chart a path toward more robust, generalizable, and socially responsible VAD.

## 2 RELATED WORK

**Surveys and algorithmic taxonomies.** A number of surveys organize the VAD literature by models and paradigms (*e.g.*, reconstruction-based models (Liu et al., 2018), memory-augmented architectures (Rossi et al., 2024), contrastive and self-supervised methods (Georgescu et al., 2021), and

weakly-supervised/video-level approaches (Wu et al., 2024)). These surveys summarize important benchmarks (Vijay et al., 2010; Lu et al., 2013; Luo et al., 2017; Sultani et al., 2018b; Wu et al., 2020; Ramachandra & Jones, 2020; Acsintoae et al., 2022), and provide useful maps of algorithmic progress. However, their treatment of datasets is largely descriptive: datasets are reported by size, domain, or label type, but rarely analyzed as design artifacts with systematic consequences for what models can learn. In contrast, our work treats datasets themselves as first-class objects and asks how explicit and implicit design choices shape the hypothesis class accessible to learning algorithms.

**Supervision regimes and label density.** Dataset annotation granularity strongly conditions the goal and difficulty of VAD: pixel- and frame-level benchmarks enable localization and temporal boundary estimation, while segment- or video-level benchmarks push models toward detection and retrieval without fine localization (Gong et al., 2019; Sultani et al., 2018a). Synthetic or semi-synthetic constructions (*e.g.*, datasets that artificially insert anomalies or separate anomaly types across splits) have been used to examine generalization under controlled shifts. Prior studies note these differences, but do not unify them into a principled, measurable factor (Kiran et al., 2018). We make *temporal labeling density* explicit in our framework and analyze how annotation granularity interacts with content, distributional properties, and evaluation practices to produce confounding effects in cross-dataset comparison.

**Modality, context, and human involvement.** Most classical VAD benchmarks rely on monocular RGB surveillance footage captured from fixed cameras; however, exceptions exist that broaden the sensing space (*e.g.*, XD-Violence with audio, CADP (Shah et al., 2018) and traffic-focused sets emphasizing vehicles, CUVA (Du et al., 2024) with multimodal textual ties, or synthetic datasets like UBnormal that trade identity risk for experimental control). Equally important is the dimension of human involvement: many surveillance datasets expose faces, gait, and identity cues, raising privacy and ethical considerations that influence both dataset design and downstream model behavior Zhu et al. (2024). Prior work typically discusses modality, context, and privacy in isolation (Liu et al., 2025; Samaila et al., 2024); our framework integrates these aspects under unified axes (*e.g.*, content and societal impact) and quantifies imbalance across benchmarks.

**Cross-dataset evaluation, distribution shift, and generalization.** Cross-scene and cross-dataset evaluations have occasionally appeared in the literature (Aich et al., 2023), but these efforts rarely disentangle which dataset factors (*e.g.*, taxonomy, modality, annotation density, *etc.*) drive the observed performance changes. As a result, failure modes remain ambiguous and conclusions about generalization are weak. The broader ML literature on domain generalization and dataset shift provides principled protocols for separating source and target environments (Zhou et al., 2022; Wang et al., 2022), but VAD has lacked comparable analyses that connect performance degradation to concrete, auditable dataset properties (Zhu et al., 2024). By profiling benchmarks along shared dimensions, our approach makes the nature of cross-dataset shifts explicit and enables reproducible, controlled cross-family evaluation.

**Benchmark construction and dataset difficulty estimation.** There have been several calls for larger, more diverse benchmarks or richer annotations in VAD, and various model-dependent difficulty measures (*e.g.*, loss-based metrics or calibration-based proxies (Menon & Williamson, 2018; Deng et al., 2022)) have been proposed. These contributions improve scale or assessment, but they stop short of producing a compact, model-agnostic structural profile of a dataset. What is often missing is a reproducible representation that summarizes how a dataset is positioned along the axes that truly matter for anomaly detection (*e.g.*, taxonomy granularity, motion complexity, modality coverage, temporal labeling, distributional breadth, and privacy exposure). Our work fills this gap by providing quantitative per-dimension profiles, two aggregate indicators that summarize breadth and ecosystem balance, and a cluster-aware evaluation protocol that isolates within-family performance gains from genuine cross-family generalization.

**Ethics, privacy, and stakes of benchmark design.** Research in privacy-preserving vision and fairness has shown that dataset choices (*e.g.*, inclusion of faces or sensitive contexts) have concrete consequences for both utility and ethical risk (Zhao et al., 2025; Fioretto et al., 2022). While many VAD benchmarks were created with operational objectives (*e.g.*, surveillance), few explicitly document or measure privacy exposure, identity leakage risk, or fairness implications. We embed societal impact as a core axis of our dimensional analysis, arguing that ethical properties of datasets must be treated as design constraints on par with content and annotation choices.

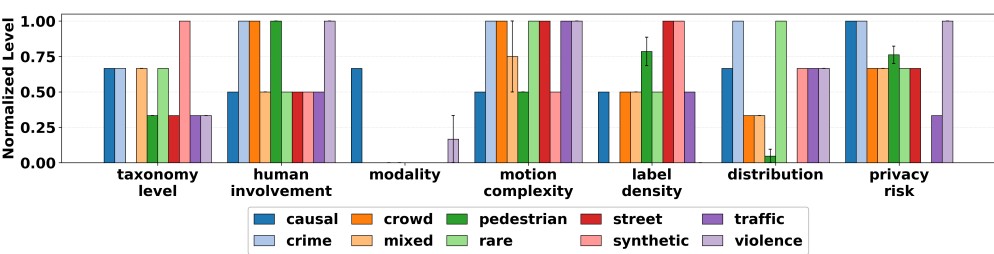

Figure 2: Theme fingerprints over seven design dimensions. Each bar shows the normalized centroid ([0, 1]) of a theme within a dimension; error bars mark standard errors across datasets in that theme.

**Relationship to broader critiques of benchmarks.** Critiques of canonical benchmarks in computer vision and NLP (*e.g.*, concerns raised about representativeness, dataset documentation, and benchmark-driven research behaviour) emphasize that benchmarks do not merely evaluate progress, they actively shape it (Ethayarajh & Jurafsky, 2020; DeYoung et al., 2019; Eriksson et al., 2025). Our dimensional approach operationalizes that critique for VAD: rather than only cataloging models or assembling larger datasets, we produce an auditable characterization of dataset structure that can be used to interpret past results and guide future benchmark construction.

To our knowledge, this is the first work to present a systematic, multi-axis framework that: (i) produces quantitative profiles for widely used VAD benchmarks, (ii) aggregates these profiles into concise indicators of coverage and balance, and (iii) uses cluster-aware, cross-family evaluation to show how dataset structure drives rank volatility and generalization failures. These elements convert qualitative observations about dataset bias into actionable, measurable procedures for evaluation and benchmark design. The remainder of the paper demonstrates the framework on a curated set of 18 benchmarks, analyzes the structural imbalances uncovered, and derives concrete design principles for next-generation VAD datasets.

## 3 DATASET STRUCTURE: FRAMEWORK AND METRICS

We operationalize our critique of hidden assumptions by introducing a *seven-dimension profiling framework* that converts dataset descriptions into auditable, quantitative representations: anomaly taxonomy, human involvement, sensing modality, motion complexity, temporal labeling density, distribution, and privacy implications (see Fig. 1).

**Seven-dimension profiling framework.** We provide a set of concrete artifacts that transform qualitative dataset descriptions into quantitative vectors. These include: (i) a scoring manual with explicit rules for each of the seven dimensions, (ii) a fixed mapping from discrete labels to numeric levels, (iii) a dataset-by-dimension matrix with version tags, and (iv) a logging file recording the specific evidence supporting each label. Fig. 1a visualizes the matrix as a clustered heatmap, showing coherent dataset families that recur throughout our analyses. A compact *pedestrian-corridor* family includes UCSD Ped1/2 (Wang & Miao, 2010), CUHK Avenue (Lu et al., 2013), SubwayEntrance (Adam et al., 2008), and SubwayExit (Adam et al., 2008). A broader *campus-and-street* family covers ShanghaiTech (Luo et al., 2017), IITB-Corridor (Rodrigues et al., 2020), Street Scene (Ramachandra & Jones, 2020), NWPU Campus (Cao et al., 2023), and MSAD (Zhu et al., 2024). A *violence* family contains RWF-2000 (Cheng et al., 2021) and XD-Violence (Wu et al., 2020), while a *multi-source long-tail* family groups UCF-Crime (Sultani et al., 2018b) and RareAnom (Thakare et al., 2023). UBnormal (Acsintoae et al., 2022) and CUVA (Du et al., 2024) form a mechanism-special pair. A complementary t-SNE projection (Fig. 1b) confirms these structural clusters: corridor datasets cluster tightly, campus-and-street spreads nearby, violence and long-tail datasets occupy separate regions, and UBnormal, CUVA, and UMN emerge as mechanism outliers.

**Quantifying ecosystem skew.** To make biases in the VAD benchmark ecosystem explicit, we compute distributional statistics. For each dimension, we calculate level frequencies and Shannon entropy, while estimating uncertainty using percentile intervals obtained via bootstrap resampling. Fig. 1d concretely illustrates these skews. Sixteen out of eighteen datasets rely solely on RGB video; only XD-Violence includes audio and only CUVA provides aligned textual modalities. For anomaly taxonomy, 11 datasets use limited categories, 5 adopt diverse labels, and single-category

or open-set taxonomies appear only once each. Segment-level supervision is rare (3/18), with the remainder providing frame- or pixel-level annotations. The dataset distribution favors single place collections, with few multi-place, multi-type, and privacy exposure is mostly medium to high. We further examine structural correlations via pairwise chi-square tests, reporting effect sizes. For example, segment-level supervision is more likely to co-occur with multi-source long-tail or violence datasets, and strong human involvement correlates with higher privacy exposure. These model-agnostic statistics show that current evaluation largely occupies a narrow slice of the anomaly detection design space. Importantly, these findings directly motivate our cross-family evaluation protocol and inform data-augmentation priorities discussed in Sec. 5.

**Family assignment and the outlier catalogue.** We define *families* as data-driven clusters derived from the seven-dimension profiles, while *themes* are human-readable tags for interpretability. Families serve as the structural units throughout our analyses. Importantly, families are not artifacts of a single linkage method: in the seven-dimensional standardized space, we compute family centroids and measure the ratio of mean inter-centroid distance to mean within-family variance, which is significantly higher than expected from random partitions of the same size. We also repeat clustering under multiple distance metrics (*e.g.*, Euclidean, cosine) and linkage methods (*e.g.*, average, complete), computing adjusted Rand index and pairwise consistency across runs. High consistency confirms that the families reflect genuine structural groupings rather than arbitrary clustering choices. Fig. 1b shows a t-SNE projection of the same profiles, labeling each point by dataset and coloring by theme. The *pedestrian-corridor* datasets cluster tightly in the lower-right region; the *campus-and-street* sets (ShanghaiTech, IITB-Corridor, Street Scene, NWPU Campus, MSAD) form a broader band nearby; *violence* (RWF-2000, XD-Violence) and *long-tail* (UCF-Crime, RareAnom) datasets occupy distinct upper-right regions. UBnormal and CUVA stand apart as mechanism-special outliers due to open-set supervision and RGB+text alignment, while UMN is isolated as a high-motion, single-mechanism crowd case. We also provide an *outlier catalogue* to systematically report datasets that deviate from family norms. Each dataset receives an *outlier index*, defined as the Mahalanobis distance to its nearest family centroid in the standardized space; an analogous definition is applied in the embedded space for geometric intuition. We additionally define *bridge sets* as datasets with similar distances to two centroids, where small ratios indicate cross-family bridging behavior. Full rankings and stability tables are provided in the appendix.

**Per-dataset indicators: coverage and balance.** To summarize each dataset's seven-dimension profile into actionable metrics, we introduce two per-dataset indicators: *coverage* and *balance*. *Coverage* measures how broadly a dataset spans the design space. Each dimension is mapped to discrete levels $\{1,2,3,4\}$, and the number of unique levels $U$ present across all seven dimensions is counted. Coverage is then computed as Coverage $= U/4 \in [0,1]$, with higher values indicating engagement across multiple tiers of the design space. *Balance* quantifies how evenly a dataset distributes its emphasis across levels. We aggregate all seven dimensions into a histogram over levels 1-4, compute Shannon entropy, and normalize by $\log_2 4$ to produce Balance $\in [0,1]$. High Balance reflects a dataset that spreads cues evenly across tiers, while low Balance indicates concentration on a narrow subset of levels. For ecosystem-level analysis, we compute normalized entropy per dimension across datasets and average over dimensions to quantify systemic skew. Fig. 1c plots Coverage against Balance for each dataset, with colors and markers following the family/theme coding introduced earlier. The dashed curve shows a fitted upper envelope, and the panel reports the Pearson correlation. Datasets with broad tier engagement and even distribution (*e.g.*, UBnormal, SubwayEntrance/Exit, XD-Violence, RWF-2000, RareAnom) occupy the upper-right quadrant, while narrow or uneven datasets (*e.g.*, CADP, MSAD) lie in the lower-left, with intermediate datasets (*e.g.*, CUVA, NWPU Campus) filling the middle. These indicators serve complementary operational roles. Coverage functions as a *planning signal*: datasets with higher Coverage offer broader structural exposure, providing an upper envelope for cross-family baseline performance. Balance acts as a *stability signal*: datasets with lower Balance are prone to higher score volatility, and in Sec. 5, we show that low Balance correlates with increased cross-model variance and rank instability on the intersection of six benchmarks. Confidence intervals and sensitivity analyses are reported in the appendix. Practically, Coverage informs selection of minimal yet representative training or evaluation subsets, while Balance guides the prioritization of dimensions to reduce evaluation instability and identify the most informative benchmarks for challenging generalization scenarios.

## 4 DATASET PROFILING AND STRUCTURAL INSIGHTS

We examine the benchmark ecosystem by profiling mechanism families, quantifying dimensional co-occurrence, and highlighting outliers and bridges that stress-test generalization.

**Family profiles and representative mechanisms.** We summarize dataset *mechanism families* by computing theme centroids in the seven-dimensional design space and visualizing them as interpretable *fingerprints*. Each centroid is derived from the dataset-by-dimension matrix using the fixed mapping and scaling protocol, with values linearly normalized to $[0, 1]$ within each dimension to ensure comparability across axes. Error bars indicate the standard error across datasets sharing the same theme. This transformation converts qualitative theme labels into quantitative, comparable profiles, revealing both dominant dimensions and bridging factors across families. Fig. 2 shows these fingerprints. The pedestrian theme exhibits high human involvement and dense labeling while remaining minimal on modality, reflecting people-centric, tightly supervised scenes. Street-themed datasets elevate motion complexity and domain-constrained distribution, capturing mixed agent dynamics in fixed RGB footage. Crime datasets show increased distribution breadth and privacy exposure, representing multi-source recordings with identity-revealing content, whereas Rare datasets emphasize long-tail distribution, highlighting low-frequency anomalies. Crowd datasets are distinguished by high motion with comparatively limited taxonomy, indicative of flow-dominated behavior. Violence datasets peak simultaneously on human involvement and motion and show non-zero modality when audio contributes, exposing action-sound coupling. Traffic datasets increase motion and distribution while exhibiting mixed human involvement, acting as a bridge between street and agent-centric regimes. Synthetic datasets sharply reduce privacy risk while preserving high label density, isolating supervision effects without identity exposure. Causal datasets elevate taxonomy and modality via text alignment, emphasizing reasoning beyond single-stream RGB, while mixed datasets maintain a broadly balanced profile, serving as a neutral reference. Interpreted alongside Fig. 1, these fingerprints reveal that the dominant axes separating families are human involvement paired with label density, motion with distribution, and modality with privacy. They also indicate which dimensions should be adjusted when evaluating or claiming generalization across mechanism families, providing actionable guidance for both dataset design and cross-family evaluation.

**Co-occurrence and causal hints from the dimensions.** Systematic co-occurrence among dataset design dimensions can introduce unintended shortcuts for models. For example, narrow anomaly taxonomies paired with human-centric footage may cause detectors to conflate identifiability with abnormality. Similarly, long-tail datasets combined with coarse segment-level labels and high motion can bias models toward coarse temporal predictions with uncertain boundaries. Robust evaluation therefore requires breaking such co-occurring patterns to prevent inflated within-family performance from being misinterpreted as genuine generalization. Fig. 3 quantifies these associations using Cramér's $V$ on contingency tables derived from the level counts of each dimension. The strongest association occurs between taxonomy and human involvement ($V = 0.66$), while taxonomy also aligns significantly with privacy and distribution (0.60 and 0.55, respectively). Distribution co-occurs with both motion and labeling density ($V = 0.56$), reflecting that multi-source long-tail sets often involve higher motion and sparser, segment-level supervision. Human involve-

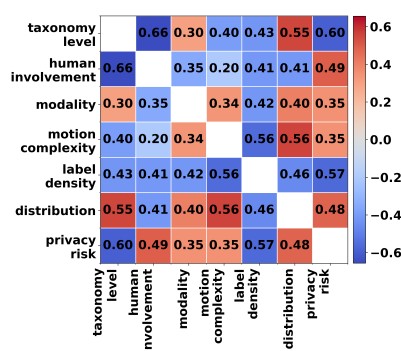

Figure 3: Pairwise co-occurrence among the seven design dimensions. Each cells shows Cramér's $V$ (range $[0, 1]$), with larger values indicating stronger associations. Numerical values are overlaid for clarity.

ment and privacy remain substantially correlated ($V = 0.49$), and moderate associations between modality and both labeling density and distribution (0.40-0.42) indicate that additional modalities such as audio or text are primarily introduced in datasets with relaxed supervision. These quantified associations directly motivate controlled evaluation protocols. Training on RGB-only, limited-taxonomy families while testing on families with broader modality or distribution diversity explicitly breaks both cue availability and class coverage. Conversely, training on segment-level long-tail datasets and testing on single-domain, frame- or pixel-labeled sets reverses supervision granularity

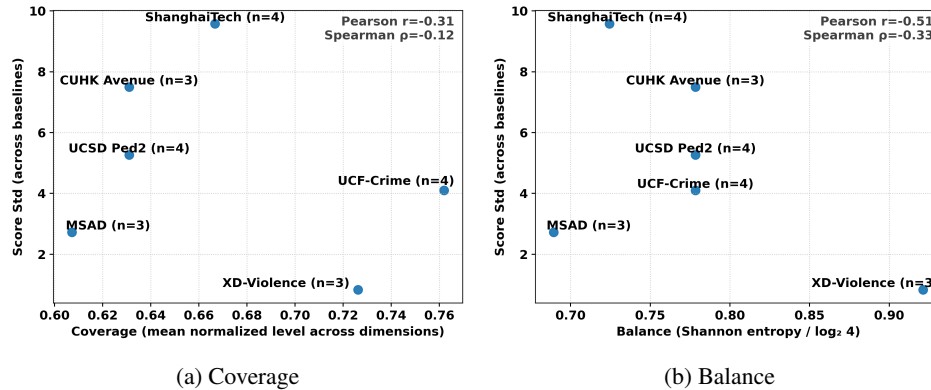

(a) Coverage                (b) Balance

Figure 4: Structural indicators *vs*. cross-model variability. Each point is a benchmark; the $y$-axis shows the AUC standard deviation across four baselines ($n$ in labels = number of models). (a) Coverage shows a weak negative correlation with variability, limited by its narrow range on this subset. (b) Balance shows a stronger negative correlation: less-balanced datasets yield more volatile scores. Together, these results motivate reporting structure-aware volatility alongside headline metrics.

and distributional patterns. Claims of generalization should be validated across both splits to ensure that performance is not driven by structural shortcuts embedded in the benchmarks.

**Outliers and bridges.** Outlier and bridge datasets highlight regions of the design space where mechanisms shift or domains intersect, providing a cost-effective way to stress-test model extrapolation without collecting new data. We score each dataset in the standardized seven-dimensional space and identify those that either lie far from any family centroid or sit between two centroids at comparable distances. These datasets expose the largest mechanism gaps and are disproportionately informative for detecting shortcut reliance and sensitivity to distributional shifts. Fig. 5 ranks datasets by an outlier index in the z-scored profile space. Three datasets emerge as prominent structural outliers: CUVA (4.43), due to RGB+Text alignment enabling causal reasoning; UBnormal (3.63), because of open-set supervision with pixel-level masks; and UMN (2.79), representing a high-motion, single-mechanism crowd scenario. A secondary tier includes Street Scene (2.51) and CADP (2.21), functioning as

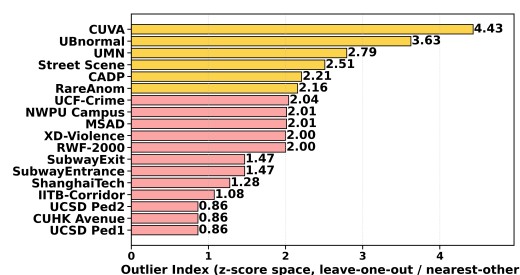

Figure 5: Top structural outliers and bridge datasets. Bars show outlier scores in the standardized seven-dimensional space (leave-one-out, nearest-other formulation; equivalent ordering to a Mahalanobis index). Yellow marks highlight atypical or bridging datasets (*e.g*., CUVA, UBnormal, UMN, Street Scene, *etc*.), while pink marks denote family-core sets. Outliers and bridges are recommended as stress tests and should be reported with rank-instability metrics.

bridges toward street and traffic mechanisms, along with RareAnom (2.16), which introduces longtail rarity. Long-tail crime datasets (UCF-Crime, 2.04) and violence datasets (RWF-2000, XD-Violence, 2.00) sit above their family cores, whereas corridor sets (UCSD Ped1/Ped2, Avenue) remain near the baseline (0.86-1.08). These values justify a concrete reporting protocol: claims of generalization should explicitly include performance on outlier and bridge datasets and document rank instability when evaluation is restricted to these items. Without such stress testing, strong within-family results can mask failures precisely in the scenarios where mechanisms shift or domains intersect, undermining claims of robust generalization.

# 5 STRUCTURE-GUIDED EVALUATION

Benchmark scores can vary widely depending on dataset composition and structural properties. To understand how the design of VAD datasets influences leaderboard stability and cross-model com-

parability, we analyze publicly available results through the lens of the previously defined structural indicators: Coverage and Balance.

**Coverage and balance as predictors of cross-model variability.** To quantify how dataset structure influences leaderboard stability, we examine six widely reported benchmarks with overlapping public results: UCF-Crime, ShanghaiTech, UCSD Ped2, CUHK Avenue, XD-Violence, and MSAD. For each dataset, we collect AUC scores from four representative baselines (RTFM (Tian et al., 2021), MGFN (Chen et al., 2023b), TEVAD (Chen et al., 2023a), and EGO (Ding et al., 2025)). Dataset-wise variability is computed as the standard deviation across available baselines, with missing entries treated as NaN and excluded from the calculation (*e.g.*, TEVAD lacks CUHK Avenue and MSAD; EGO lacks XD-Violence). Both Pearson and Spearman correlations are computed to quantify monotonic and linear relationships. Fig. 4 presents the results. Coverage exhibits only a weak negative correlation with cross-model variability on this six-dataset subset (Fig. 4a; Pearson $r = -0.31$, Spearman $\rho = -0.12$), reflecting the limited spread of Coverage levels in this intersection. In contrast, Balance is more predictive (Fig. 4b; $r = -0.51$, $\rho = -0.33$): datasets with lower Balance, *i.e.*, uneven distribution of structural cues, show higher cross-model variance and consequently more unstable leaderboard rankings. This finding carries a practical implication for benchmarking: in addition to reporting standard performance metrics such as AUC, authors should provide each dataset's Balance along with the observed cross-model standard deviation. Doing so makes the stability of method comparisons explicit, highlighting which datasets amplify uncertainty and which offer reliable signals for evaluating generalization.

**Dataset-wise variability as a measure of discriminative power.** Building on the baseline performance matrix, we quantify each dataset's ability to differentiate methods under identical evaluation conditions. For every dataset, we compute the cross-model standard deviation of AUC scores (without imputation), treating this value as a dataset-centric measure of *discriminative power*. Unlike aggregate leaderboard rankings, which conflate performance across multiple datasets, this metric reveals which benchmarks provide the strongest signal for separating models. Fig. 6 shows the ranking of datasets by discriminative power. ShanghaiTech exhibits the highest spread (std = 9.6 AUC points, $n = 4$), followed by CUHK Avenue (7.5, $n = 3$) and UCSD Ped2 (5.2, $n = 4$). UCF-Crime shows moderate variability (4.1, $n = 4$), MSAD lower (2.8, $n = 3$), and XD-Violence demonstrates the greatest agreement across methods (0.9, $n = 3$). These results provide concrete guidance for benchmarking: when evaluating generalization under fixed computational or data budgets, at least one high-variability suite (*e.g.*, ShanghaiTech or CUHK Avenue) should be included, and dataset-wise variability should be re-

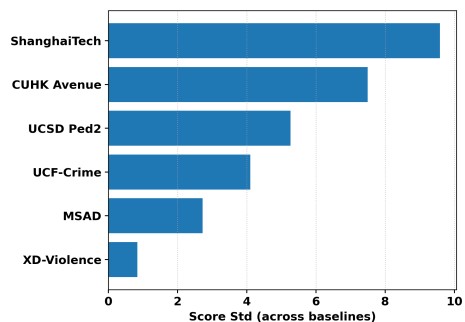

Figure 6: Dataset-wise variability as a measure of discriminative power. Bars show the standard deviation of AUC across baselines (higher = more discriminative). ShanghaiTech and CUHK Avenue induce the largest spread, UCSD Ped2 is moderate, and XD-Violence yields the most consistent scores. Reporting this variability highlights benchmarks that better separate methods and complements single-number leaderboards.

ported explicitly. This ensures that rank sensitivity is transparent and that differences in model performance are interpretable rather than obscured by low-variance datasets.

**Rank-based instability under structural shifts.** To capture how dataset structure affects leaderboard consistency, we analyze rank stability across datasets using order statistics, which avoids scale effects inherent in raw AUC values. For the six benchmarks with overlapping results (UCF-Crime, ShanghaiTech, UCSD Ped2, CUHK Avenue, XD-Violence, MSAD) and four baselines (RTFM, MGFN, TEVAD, EGO), we first convert per-dataset AUC scores into model ranks, with higher AUC corresponding to higher rank. For every dataset pair, we compute Kendall's $\tau$ using only shared models and leave pairs with fewer than three shared models as missing. The resulting $6 \times 6$ rank correlation matrix directly quantifies how consistently two datasets order the same set of methods. We also report, for each dataset, the mean Kendall's $\tau$ with all other datasets, serving as a dataset-level indicator of rank stability. Fig. 7 shows key differences. XD-Violence shows the highest cross-dataset agreement (mean $\tau = 0.78$), followed by ShanghaiTech (0.60) and UCF-Crime /

UCSD Ped2 (both 0.53). MSAD exhibits moderate stability (0.50), whereas CUHK Avenue shows near-zero agreement (0.00), indicating frequent rank inversions across suites. Cells with fewer than three shared models are masked to avoid spurious extremes due to insufficient overlap. Combined with the dataset-wise variance analysis, these results demonstrate that low cross-model variance does not guarantee stable ordering, and datasets with low mean $\tau$ should be treated as potential sources of leaderboard instability.

**Rank volatility as a reportable metric.** To make leaderboard stability auditable, we distill cross-model variability into two complementary metrics. For each dataset, we define $\sigma_{\mathrm{auc}}$ as the standard deviation of AUC across all available baselines (no imputation; NaNs excluded) and $\bar{\tau}$ as the mean Kendall's $\tau$ with all other datasets, computed only on shared models with a minimum of three (off-diagonal entries; NaNs ignored). $\sigma_{\mathrm{auc}}$ quantifies how strongly a dataset discriminates between methods, while $\bar{\tau}$ captures the consistency of model rankings across datasets. Applying this to the six-dataset intersection with four baselines (RTFM, MGFN, TEVAD, EGO), we find that datasets with high $\sigma_{\mathrm{auc}}$ (*e.g.*, ShanghaiTech, CUHK Avenue) provide strong separation between methods, making them valuable for comparative evalu-

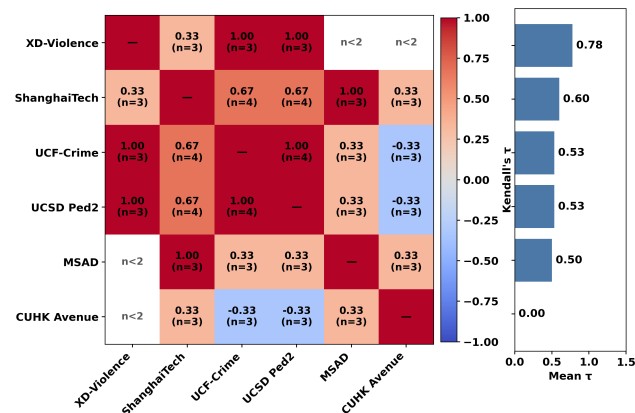

Figure 7: Rank consistency across datasets (Kendall's $\tau$). Pairwise $\tau$ values compare dataset-specific model rankings (higher = more consistent). Computed on shared models only; pairs with fewer than three shared models are omitted. Right panel: mean $\tau$ per dataset (off-diagonal, NaNs ignored). Model set = {RTFM, MGFN, TEVAD, EGO}.

ation. In contrast, datasets with low $\bar{\tau}$ (*e.g.*, CUHK Avenue) exhibit frequent rank inversions across suites, highlighting instability under structural shifts. We recommend reporting, alongside standard metrics like AUC or AP, the tuple $(\sigma_{\mathrm{auc}}, \bar{\tau}, n)$ per test set, where $n$ denotes the number of baselines contributing to $\sigma_{\mathrm{auc}}$. This practice allows reproducible assessment of both discriminative power and ranking stability, enabling transparent and auditable comparisons without additional training.

## 6 CONCLUSION

This paper establishes **a principled framework** to make VAD benchmarks measurable, auditable, and reproducible. By formalizing datasets as seven-dimensional profiles, releasing a public dataset-by-dimension matrix, and introducing two per-dataset indicators: Coverage (unique-level fraction across dimensions) and Balance (normalized entropy of level frequencies), we provide both a structural and quantitative lens on benchmark design. Our analysis delivers **several concrete insights**. Current benchmarks exhibit strong skew in modality and anomaly taxonomy, and stable families emerge across clustering methods. Outlier and bridge datasets highlight mechanism transitions, revealing where models are most likely to fail or exploit shortcuts. Structure predicts variability: datasets with low Balance exhibit higher cross-model AUC variance, while $\sigma_{\mathrm{auc}}$ identifies datasets most discriminative for comparing methods, and mean Kendall's $\tau$ exposes rank instability across suites. These results make explicit how dataset design directly impacts model evaluation and perceived generalization. From these findings, we derive **actionable guidelines** for the community. Researchers should report per-dataset $(\sigma_{\mathrm{auc}}, \bar{\tau}, n)$ alongside headline AUC or AP to qualify leaderboard stability. Claims of cross-family generalization must include at least one high-variability suite and explicit outlier/bridge datasets to stress-test extrapolation. Dataset selection should prioritize high Coverage and balanced profiles to avoid evaluating models in restricted or biased slices of the design space. For **future benchmark design**, skew analysis suggests adding modalities beyond RGB, expanding anomaly taxonomy, and incorporating multi-source or long-tail data to improve structural coverage. By making structure measurable and evaluation stability reportable, this framework provides a practical path toward broader coverage, fairer comparisons, and reproducible, socially responsible evaluation in VAD.

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

## A  APPENDIX

### A.1  DATASET OVERVIEW AND METADATA.

#### A.1.1  DATASET OVERVIEW

This section provides detailed descriptions of the 18 VAD datasets included in our dimensional analysis. Each entry summarizes the dataset's structure, anomaly types, source modality, annotation granularity, and any privacy or human bias concerns, serving as a reference for both our dimensional scoring and model evaluation.

**SubwayEntrance & SubwayExit** consist of long-duration surveillance footage from fixed CCTV cameras overlooking subway turnstile corridors, one facing the entrance and one facing the exit. The scenes exhibit strong temporal autocorrelation and distinct rush-hour periodicity, which makes evaluation sensitive to how training and test segments are sliced. Anomalies include walking in the wrong direction through one-way gates, fare evasion by jumping barriers, loitering midstream, and brief crowding near the turnstiles. Most studies follow an unsupervised or weakly supervised protocol, training only on early normal segments and evaluating on the full sequence using frame-level AUC. A few later works also explore spatial localization, though unlike UCSD or Avenue, no official pixel-level masks are provided. Common preprocessing steps include grayscale conversion, temporal subsampling, and cropping a region of interest around the gates to suppress static background clutter. Due to the prominent presence of people and visible body movements, these sequences introduce strong human-centric biases and significant privacy exposure. Models may overfit to pedestrian motion patterns instead of learning generalizable mechanisms behind anomalies. Issues like perspective-induced distortion, frequent partial occlusions at the gates, and extremely sparse anomalies further complicate training and evaluation.

**UMN** contains three short video sequences staged in controlled environments: two outdoor scenes and one indoor hall. Each clip begins with normal crowd behavior and transitions abruptly into collective panic and dispersal. All videos are low-resolution but are designed to create a clear global shift in group dynamics. Annotations are typically frame-level, with the abnormal segment beginning at the scripted panic onset. Most studies train on the normal portion and evaluate over the full video. UMN is particularly useful for evaluating methods that model holistic motion patterns, such as optical flow consistency, low-rank and sparse decompositions, or physical crowd simulations. While the simplicity and clarity of the anomaly make UMN a good benchmark for verifying global behavioral sensitivity, its lack of environmental diversity, scripted transitions, and cooperative actors may lead to inflated separability when compared with in-the-wild surveillance footage. It remains valuable as a sanity check for testing models under global motion regime shifts.

**UCSD Ped1 & Ped2** are two widely used datasets that capture pedestrian walkways on a university campus, with contrasting camera perspectives and persistent small-object challenges. Ped1 is recorded facing along the direction of motion, leading to strong perspective distortion: distant objects are extremely small and quickly enlarge as they approach. Ped2 uses a lateral view with more consistent object scale across the field of view. The official splits include 34 train / 36 test videos for Ped1, and 16 train / 12 test for Ped2. Annotations are available at the frame level, and a subset of test videos includes pixel-level masks for spatial localization. Anomalies typically involve non-pedestrian entities entering pedestrian zones, such as bicycles, carts, or skateboarders. Unusual motion patterns also appear occasionally. Challenges include severe class imbalance due to sparse anomalies, small distant objects in Ped1 that are difficult to detect, and visual ambiguities caused by shadows and reflections. Both datasets favor methods that can extract fine-grained motion against a backdrop of regular pedestrian flow and can handle perspective-induced scale variations. Since footage often reveals identifiable silhouettes, privacy concerns should be considered in any public visualizations.

**CUHK Avenue** captures pedestrian activity in a campus walkway using a fixed outdoor camera. The dataset contains 16 training and 21 testing videos, totaling 30,652 frames (15,328 for training and 15,324 for testing). Annotations include frame-level labels and, for many videos, corresponding pixel-level masks available from the dataset's project site. This combination of detection and localization made Avenue one of the earliest benchmarks to encourage both types of evaluation. Anomalies are typically localized and motion-driven rather than semantically diverse. Examples include throwing objects into the walkway, abrupt running, sudden stops, and isolated gestures. These

properties make Avenue suitable for models that emphasize motion magnitude, flow consistency, and trajectory regularity. Methods relying heavily on semantic diversity may under-perform due to the narrow anomaly taxonomy. Compared with UCSD Ped1 and Ped2, Avenue has more consistently sized targets with fewer extreme perspective distortions, which helps with localization performance. However, the similarity among anomaly types and the short bursts of motion against otherwise slow pedestrian flow can lead to performance saturation for models tuned to motion saliency. Practical issues include intermittent shadows, illumination fluctuations, and mild occlusions near benches or lampposts. Spatial cropping is less common here due to the wider field of view. As the footage clearly shows people's faces and bodies, privacy concerns are relevant when visualizing results.

**ShanghaiTech** extends the campus surveillance paradigm to a larger and more diverse setting. It consists of 437 video clips collected across 13 distinct scenes, with 330 used for training and 107 for testing. In the commonly used unsupervised protocol, models are trained on normal-only clips and evaluated on test videos containing anomalies. Frame-level labels are provided to support detection metrics. ShanghaiTech is designed to probe cross-scene generalization by introducing variation in crowd density, background layout, and camera placement. A single type of behavior, such as cycling, can appear very differently across scenes due to changes in scale, perspective, and motion context. These variations require models to adopt robust normalization, temporal aggregation, and adaptation mechanisms. Patch-based autoencoders and memory-augmented networks often benefit from scene-specific calibration, while methods that generalize across settings must avoid overfitting to camera geometry. Reported challenges include complex shadows cast by buildings or trees, slight camera jitter, and inconsistent motion parallax that weakens background subtraction techniques. Because anomalous events are relatively sparse and vary significantly across locations, careful threshold calibration and cross-scene consistency are critical. Researchers typically report both per-scene and overall metrics, and qualitative localization is often visualized using masks available from community re-annotations. Similar to other pedestrian-centric datasets, ShanghaiTech raises human-centric bias and privacy concerns due to the prominent presence of people in public spaces.

**UCF-Crime** is a large-scale benchmark designed for weakly supervised anomaly detection in real-world surveillance footage. It contains 1,900 untrimmed videos totaling approximately 128 hours, spanning 13 anomaly categories such as fighting, robbery, arson, accidents, and burglary. Labels are provided at the video level, indicating the presence or absence of an anomaly within each clip, and a separate protocol also treats it as a multi-class event classification task. The dataset emphasizes the need for long-range temporal reasoning under weak supervision. Anomalous events typically occupy only a small portion of each video, requiring models to suppress large volumes of irrelevant content while detecting short, high-salience episodes. As the dataset does not include audio, all methods rely solely on visual information. This constraint encourages architectures that employ temporal attention, top-k pooling, or ranking losses based on multiple-instance learning frameworks. The diversity of scenes introduces considerable domain variation, including different lighting conditions, motion blur, compression artifacts, and varying levels of occlusion. Label noise is also present, as some videos labeled "normal" may still contain visually abnormal content. These factors make UCF-Crime a demanding testbed for robustness and generalization. Evaluation is usually conducted using video-level AUC, with optional segment-level diagnostics based on proposal scoring. Due to the real-world nature of the footage and the visibility of people and incidents, ethical considerations such as identity masking and responsible usage are essential in modern evaluation pipelines.

**CADP** focuses on traffic accidents and near-miss events collected from online sources, aiming to bridge the gap between curated driving datasets and uncontrolled roadside surveillance. It includes 1,416 video clips, among which 205 contain full spatiotemporal annotations with bounding boxes and event timestamps. These annotated clips support dense supervision for detection, tracking, and temporal localization. The remaining clips contain weak temporal cues, making them suitable for accident anticipation or retrieval-style tasks. Visual diversity is a key feature of CADP. Camera viewpoints range from dashboard cameras and roadside poles to hand-held phones. Lighting conditions span daytime and nighttime. Compression artifacts, rolling shutter distortions, and motion blur are frequently present. Typical anomalies include multi-vehicle collisions, loss of control, red-light violations, and sudden lane intrusions. Pedestrians and cyclists appear in some videos, although they are often small, partially occluded, or only briefly visible. These properties make CADP a challenging benchmark for detecting rare motion patterns and reasoning over small, fast-moving objects. It is widely used for accident detection, impact prediction, and multi-object reasoning. Modeling pitfalls include class imbalance caused by dominant normal driving segments, ambiguous annota-

tions in multi-actor scenes, and view or domain shifts that reduce the effectiveness of single-camera heuristics. Evaluation protocols typically include segment-level mean average precision or temporal IoU, with frame-level or box-level metrics used in fully annotated clips. While privacy risks are generally lower than in pedestrian-centric datasets, identifiable elements such as license plates and faces may still be visible and should be handled carefully.

**Street Scene** is an urban anomaly detection dataset captured by a fixed daytime surveillance camera positioned above a two-lane road, a bike lane, and adjacent sidewalks. The official split includes 46 training and 35 testing sequences, collected over two summers at different times of day to capture natural environmental variation. The dataset emphasizes realistic urban scenes containing vehicles, cyclists, and pedestrians engaging in routine but diverse behaviors. Anomalies are defined based on context violations rather than appearance outliers. For example, a pedestrian walking in the bike lane or a car entering the sidewalk area is considered abnormal. These context-based anomalies require models to incorporate semantic understanding of spatial layout and behavior regularity. The dataset discourages trivial solutions that rely on imprecise blob detection by providing detailed pixel-level annotations and stricter evaluation criteria. Practical challenges include mid-ground occlusions from parked vehicles or poles, shadow dynamics caused by trees and buildings, and varying object scales across the wide field of view. Compared to earlier pedestrian-only benchmarks, Street Scene provides richer class diversity and spatial complexity, encouraging models that can reason over scene semantics, boundaries, and multi-agent interactions. Most works report both frame-level AUC and region-based overlap metrics, with qualitative maps used to assess spatial precision. Since the camera is mounted high enough to obscure fine facial detail, privacy risks are moderate, although close-range identifiable features may still appear and should be masked when necessary.

**XD-Violence** is a large-scale multi-modal dataset designed for weakly supervised detection of violence in untrimmed videos. It contains 4,754 videos totaling approximately 217 hours, with segment-level annotations indicating when violent activity occurs. The dataset covers six broad categories of violence and includes content from diverse sources such as social media and public video repositories. Unlike most prior datasets, XD-Violence provides both audio and visual modalities. RGB frames are accompanied by synchronized audio tracks, and many methods also use optical flow as a third input stream. This multi-modal structure encourages fusion strategies that integrate visual motion, acoustic cues like shouts or impact sounds, and temporal attention. Violent segments are typically brief relative to the full video length, which makes it necessary to suppress long stretches of irrelevant background content. Modeling challenges include intra-class variation in how violence manifests, temporal label imprecision, and strong domain shifts caused by diverse camera types, compression levels, and filming styles. Popular approaches include top-k pooling, multiple instance learning, and proposal scoring. Evaluation is typically conducted at the segment level using mean average precision or AUC. This aligns with the dataset's multi-modal structure, and some studies report modality ablations to examine the contribution of audio, RGB, and flow separately. Ethical concerns are significant, as some videos contain distressing or sensitive content, and faces or identifying features may be visible. Responsible usage includes applying anonymization, offering content warnings, and following clear data handling protocols.

**IITBCorridor** is an indoor surveillance dataset recorded at the Indian Institute of Technology Bombay. It features a single ceiling-mounted camera positioned to overlook a long corridor with multiple doorways. The dataset consists of 483,566 frames, providing a prolonged temporal context for modeling routine pedestrian behavior. Normal activities include walking at varying speeds, standing, and short interactions near doors. Anomalies include running, sudden falls, object dropping or throwing, and group movements that deviate from corridor norms. Many events were staged in controlled conditions to ensure adequate coverage of edge cases, including corner occlusions and group interactions. Frame-level annotations are standard, while some community adaptations introduce point-level or region-based localization tasks. The fixed viewpoint introduces strong perspective shortening toward the corridor's far end, where objects become small and challenging to detect. Reflections from polished floors and glass panels can cause transient highlights, while the narrow geometry frequently leads to partial occlusions near doors. These factors make precise localization and temporal boundary assignment difficult. Most studies report frame-level ROC or AUC metrics, and some explore trajectory-based evaluation to capture deviations from normal movement patterns. Privacy risks are present but relatively low, as the ceiling-mounted view reduces the visibility of individual facial features.

**RWF-2000** is a violence detection benchmark built from real surveillance footage. It contains 2,000 short clips, with approximately 300,000 frames in total. The dataset is designed to simulate practical public security scenarios. Each clip is a five-second segment at 30 FPS, extracted and labeled independently for binary classification. The dataset covers diverse environments, both indoor and outdoor, with variations in camera angle, lighting, and video quality. Degradations such as motion blur, compression artifacts, and camera shake are intentionally included to mimic real-world deployment conditions. Violence manifests in a wide range of actions, including pushing, striking, wrestling, and crowd surges, resulting in substantial intra-class variation. Due to the short duration of clips and the high imbalance between calm and violent frames, effective models often require temporal encoding methods that can isolate salient actions. 3D convolutional networks, optical flow backbones, and transformer-based models are frequently used. Evaluations commonly report clip-level accuracy and AUC, and some studies explore scoring sub-segments within each clip. Challenges include background-label correlation, fast viewpoint changes, and the potential for shortcut learning. Although privacy risks are moderate, identifiable faces or clothing may appear, and masking is recommended for responsible usage.

**UBnormal** is a synthetic VAD dataset built using 3D rendering engines such as Cinema4D. It enables full control over scene layout, camera trajectory, lighting conditions, and object placement. The dataset is designed to support supervised open-set anomaly detection. During training, models are exposed to pixel-level masks for a limited set of anomaly types. During testing, entirely new anomaly categories are introduced to evaluate generalization to unseen events. This separation between training and test anomalies forces models to rely on low-level motion and interaction patterns rather than memorizing specific categories. The dataset includes official splits and utilities for computing both frame-level and region-level evaluation metrics. Many studies also report region-based detection criteria such as RBDC and TBDC, which assess spatial coverage and temporal boundary accuracy. UBnormal removes real-world privacy concerns, as all entities are synthetic. This allows for dense and accurate annotation at scale. However, the domain gap between synthetic and natural footage can be substantial. Successful models often incorporate domain adaptation strategies, motion-based representation learning, or normalization techniques that reduce reliance on appearance features. The dataset is well suited for testing the effectiveness of dense supervision, evaluating localization fidelity, and analyzing generalization under controlled conditions.

**NWPU Campus** is a large-scale benchmark designed for semi-supervised anomaly detection and video anomaly anticipation. It includes 43 distinct scenes and 28 anomaly categories, covering approximately 16 hours of footage. A central design feature is scene-dependent labeling, where the same behavior may be considered normal in one location but abnormal in another. For example, riding a bicycle in a bike lane is normal, while riding in a pedestrian corridor is abnormal. This context-sensitive setup encourages models to incorporate scene semantics and spatial rules, rather than relying solely on action templates. The dataset provides both frame-level abnormality labels for detection tasks and temporally aligned annotations for anticipation tasks. It presents a wide range of challenges, including variable crowd densities, occlusions near structural elements like pillars, and lighting changes due to weather. Cross-scene generalization is a core difficulty, as identical actions may appear differently across environments. Evaluation protocols commonly report frame-level AUC, anticipation accuracy at multiple lead times, and per-scene breakdowns to assess consistency. Since people are primary actors and appear frequently, privacy and human-centric bias must be considered when visualizing results.

**RareAnom** focuses on underrepresented anomaly categories that are seldom found in traditional surveillance benchmarks. It consists of approximately 2,200 videos covering 17 rare anomaly types, collected from a wide variety of sources including CCTV cameras, dashcams, hand-held devices, and mobile phones. Each video is annotated with precise temporal boundaries to support both detection and localization. The multi-source nature of RareAnom introduces significant domain variation in resolution, compression quality, camera stability, and motion characteristics. These variations challenge models to learn anomaly mechanisms rather than source-specific artifacts. The dataset is particularly suitable for evaluating long-tail recognition, cross-device generalization, and robustness to unseen filming styles. Common strategies include long-range temporal proposal generation, loss re-weighting to address class imbalance, and domain-invariant feature extraction. Evaluation typically focuses on segment-level AUC or mean average precision, with per-category results reported to highlight performance across different rare types. Privacy concerns exist due to the presence of

| Dataset | Year | Scenes | Source | Modality | Anomaly type | Annotation Level | Human Bias | Privacy Risk |
|---|---|---|---|---|---|---|---|---|
| SubwayEntrance | 2008 | 1 | Surveillance | RGB | Pedestrian | Pixel | High | YES |
| SubwayExit | 2008 | 1 | Surveillance | RGB | Pedestrian | Pixel | High | YES |
| UMN | 2009 | 1 | Surveillance | RGB | Behavior | Frame | High | YES |
| UCSD Ped1 | 2010 | 1 | Surveillance | RGB | Pedestrian | Pixel | High | YES |
| UCSD Ped2 | 2010 | 1 | Surveillance | RGB | Pedestrian | Pixel | High | YES |
| CUHK Avenue | 2013 | 1 | Surveillance | RGB | Pedestrian | Frame | High | YES |
| ShanghaiTech | 2017 | 1 | Surveillance | RGB | Pedestrian | Frame | High | YES |
| UCF-Crime | 2018 | NA | Online Surv. | RGB | Crime | Video | High | YES |
| CADP | 2018 | 1 | Surveillance | RGB | Traffic | Frame | Low | YES |
| Street Scene | 2020 | 1 | Surveillance | RGB | Traffic | Pixel | Med | YES |
| XD-Violence | 2020 | NA | Films/Online | RGB+Audio | Violence | Segment | High | YES |
| IITBCorridor | 2020 | 1 | Surveillance | RGB | Pedestrian | Pixel | High | YES |
| RWF-2000 | 2020 | NA | Online Surv. | RGB+Optical flow | Violence | Segment | High | YES |
| UBnormal | 2022 | 8 | 3D modeling | RGB | Pedestrian | Pixel | None | NO |
| NWPU Campus | 2023 | 1 | Surveillance | RGB | Pedestrian | Frame | High | YES |
| RareAnom | 2023 | NA | Surveillance | RGB | Multiple | Segment | High | YES |
| MSAD | 2024 | 14 | Online Surv. | RGB | Multiple | Frame | Med | NO |
| CUVA | 2024 | 14 | News/Online | RGB+Text | Multiple | Segment | High | YES |

Table 1: **Metadata for all 18 analyzed datasets.** The table summarizes key dataset properties, including release year, scene diversity, data source, sensing modality, anomaly type, annotation granularity, estimated human bias, and potential privacy risk. Annotation levels are categorized as pixel-level, frame-level, segment-level, or video-level supervision, while human bias and privacy risk are estimated based on both official articles and manual inspection. This metadata forms the basis for the dimensional comparisons presented in the following sections.

identifiable individuals in some videos, and best practices include anonymization or content masking when releasing visualizations.

**MSAD** is a real-world anomaly detection dataset covering 14 different scenarios, such as streets, shopping malls, sidewalks, and parks. It combines a diverse set of camera views, lighting conditions, and motion patterns to evaluate the generalization ability of anomaly detection models across environments. Anomalies include both human-related events (such as fighting or unauthorized entry) and non-human anomalies (such as vehicles in restricted zones or hazardous object presence). The dataset offers annotations at the frame or segment level depending on task requirements. Its design encourages training strategies that perform consistently across scenarios, rather than overfitting to one specific type of scene. The same anomaly may appear quite different depending on the environment due to scale differences, occlusion patterns, and background clutter. Baseline results provided by the dataset authors make MSAD a useful reference for realistic anomaly detection tasks. Evaluation practices include reporting per-scenario and overall detection metrics, testing with and without specific scene categories, and visualizing localized failure cases. Privacy concerns are moderate. Although people appear frequently, many clips do not include clearly identifiable facial features.

**CUVA** is designed to extend VAD toward causal understanding and explanation. For each anomaly instance, the dataset includes three separate human annotations: one describes what happened (including event type, start and end time, and a concise description), the second explains why it occurred using natural language, and the third describes how it unfolded and what its consequences were. The dataset introduces a structured evaluation protocol called MMEval, which aims to assess both the correctness and explanatory quality of model outputs. CUVA supports the entire pipeline from anomaly detection to textual explanation, and it provides a prompt-based baseline along with reference code for reproduction. The videos are collected from real-world sources such as public news and online platforms, leading to diverse content but also raising ethical concerns related to identifiable faces and sensitive events. Tasks in CUVA span detection, causal attribution, and free-text explanation. Evaluation includes detection metrics, textual attribute accuracy, and both human and automatic scoring of explanations. This makes CUVA a valuable resource for studying multi-modal reasoning, temporally grounded language generation, and causal alignment in complex scenes.

### A.1.2 METADATA

In addition to the narrative descriptions provided above, Table 1 offers a structured summary of key attributes across all 18 analyzed datasets. These attributes include the release year, number of distinct scenes, data sources, sensing modalities, anomaly types, annotation granularity, estimated

human bias, and potential privacy risks. The statistics were compiled through official documentation and direct inspection of the data.

Notably, the majority of datasets are based on RGB surveillance footage and involve pedestrian or crowd-related anomalies, reflecting a persistent human-centric bias in the field. Only a few datasets incorporate additional modalities such as audio (*e.g.*, XD-Violence) or textual annotations (*e.g.*, CUVA). In terms of annotation level, there is a wide spectrum ranging from coarse video-level labels (*e.g.*, UCF-Crime) to dense pixel-level supervision (*e.g.*, UCSD, Street Scene, UBnormal).

Recent benchmarks have increasingly expanded the diversity of anomaly types and scene configurations. Datasets such as RareAnom and MSAD introduce multi-scene, multi-category settings, while CUVA and UBnormal highlight efforts to incorporate causality or synthetic control. The inclusion of privacy and human bias assessments in the table further emphasizes the ethical and methodological considerations that accompany dataset design. Together, these metadata provide a foundation for the dimensional analysis that follows.

## A.2 Dimension Scoring Protocol and Mapping

### A.2.1 Seven-Dimension Scoring Rubric

To facilitate a structured and reproducible comparison of VAD benchmarks, we define a seven-dimension scoring rubric that quantifies dataset design characteristics. These dimensions cover semantic scope, visual dynamics, supervision granularity, and privacy implications. Each dimension is assessed on a discrete ordinal scale, typically from 1 (minimal) to 3 (maximal) or from 1 to 4, based on operational definitions derived from dataset descriptions, official papers, and direct inspection. Below we describe each dimension and its scoring logic.

**Taxonomy level.** This dimension measures the semantic diversity of anomalies. 0 - Single: Only one type of anomaly (*e.g.*, panic in UMN). 1 - Limited: Small set of similar anomalies (*e.g.*, pedestrian deviation). 2 - Diverse: Multiple, semantically different anomalies (*e.g.*, traffic + person). 3 - Open-set: Large or unknown anomaly space, unseen types during testing.

**Human involvement.** This dimension quantifies how central human figures are to the anomaly. 0 - None: Humans are absent or irrelevant. 1 - Background: Humans may appear but do not cause anomalies. 2 - Shared: Anomalies may involve both humans and objects. 3 - Strong: Humans are the primary anomaly agent (*e.g.*, fighting, loitering).

**Modality.** Represents the sensory channels used: 0 - RGB only. 1 - RGB + Optical Flow. 2 - RGB + Audio. 3 - RGB + Text or multi-modal alignment.

**Motion complexity.** Refers to the spatio-temporal complexity of anomaly: 0 - Low: Static object anomalies, minimal motion (*e.g.*, parked car). 1 - Medium: Short, simple movement (*e.g.*, walking wrong way). 2 - High: Coordinated actions (*e.g.*, fighting, collisions). 3 - Extreme: Large-scale chaos or long actions (*e.g.*, panic scenes, protests).

**Label density.** Supervision granularity of anomaly labels: 0 - Video-level only. 1 - Segment-level (*e.g.*, 10-second window). 2 - Frame-level. 3 - Pixel-level (*e.g.*, segmentation masks).

**Distribution.** Assesses how broadly distributed the dataset's anomalies are: 0 - one-place, one-type: All videos are from the same camera or spot, and contain only one kind of anomaly. 1 - one-place, multi-type: All videos are from the same location, but show several different anomaly types. 2 - multi-place, one-type: Videos are recorded in different locations, but all show the same kind of anomaly. 3 - multi-place, multi-type: Videos come from different locations and contain many different kinds of anomalies.

**Privacy risk.** Assesses identifiable visual information: 0 - None: Synthetic, anonymized, or distant scenes. 1 - Low: People visible but low resolution or partial occlusion. 2 - Medium: Faces/body visible but not emphasized. 3 - High: Identity-revealing footage, *e.g.*, close-up surveillance.

To ensure consistency and transparency, each dataset was scored independently by two annotators using the above manual. Discrepancies were resolved via adjudication. No imputation was applied: missing values were kept explicit to preserve uncertainty visibility. All assignments are traceable to paragraphs in the original papers or metadata fields in the dataset release.

| Dataset | modality | taxonomy level | human involvement | motion complexity | label density | distribution | privacy risk | theme |
|---------|----------|----------------|-------------------|-------------------|---------------|--------------|--------------|-------|
| SubwayEntrance | RGB | limited | strong | medium | frame | One Place, One Type | high | pedestrian |
| SubwayExit | RGB | limited | strong | medium | frame | One Place, One Type | high | pedestrian |
| UMN | RGB | single | strong | high | frame | One Place, Many Types | medium | crowd |
| UCSD Ped1 | RGB | limited | strong | medium | pixel | One Place, One Type | medium | pedestrian |
| UCSD Ped2 | RGB | limited | strong | medium | pixel | One Place, One Type | medium | pedestrian |
| CUHK Avenue | RGB | limited | strong | medium | pixel | One Place, One Type | medium | pedestrian |
| ShanghaiTech | RGB | limited | strong | medium | pixel | One Place, Many Types | medium | pedestrian |
| UCF-Crime | RGB | diverse | strong | high | segment | Many Places, Many Types | high | crime |
| CADP | RGB | limited | mixed | high | frame | Many Places, One Type | controlled | traffic |
| Street Scene | RGB | limited | mixed | high | pixel | One Place, One Type | medium | street |
| XD-Violence | RGB+Audio | limited | strong | high | segment | Many Places, One Type | high | violence |
| IITB-Corridor | RGB | limited | strong | medium | frame | One Place, One Type | medium | pedestrian |
| RWF-2000 | RGB+Optical flow | limited | strong | high | segment | Many Places, One Type | high | violence |
| UBnormal | RGB | open-set | mixed | medium | pixel | Many Places, Many Types | low | synthetic |
| NWPU Campus | RGB | diverse | mixed | high | frame | One Place, Many Types | medium | mixed |
| RareAnom | RGB | diverse | mixed | high | frame | Many Places, Many Types | medium | rare |
| MSAD | RGB | diverse | mixed | medium | frame | One Place, Many Types | medium | mixed |
| CUVA | RGB+Text | diverse | mixed | medium | frame | Many Places, One Type | high | causal |

Table 2: **Seven-dimensional rubric with cleaned distribution labels.** Distribution column uses redefined labels based on recording locations and anomaly type variety.

### A.2.2 SCORING RESULTS AND DATASET MAPPING

Here we apply the above rubric to score all 18 datasets across the seven defined dimensions. The results are summarized in Table 2, which presents a structured mapping from descriptive dataset properties to standardized ordinal or categorical values. Each column corresponds to one of the defined dimensions, with values manually assigned by two annotators following the protocol described above.

All dimensions except for the final "theme" column are used in our subsequent quantitative analyses. Discrete scores are used for taxonomy level, human involvement, modality, motion complexity, label density, distribution type, and privacy risk. For modality, multi-modal datasets such as XD-Violence and CUVA are clearly distinguished. Distribution labels are normalized into four types based on recording diversity and anomaly variety. The "theme" column provides a high-level summary of each dataset's dominant scenario, included here for interpretability.

This structured mapping enables dataset-level comparisons and supports downstream analyses, such as clustering, dimensional diagnostics, and instability quantification across benchmarks.

### A.3 MODEL EVALUATION STATISTICS AND VARIABILITY INDICATORS

In this section, we report the performance of four widely-used baselines, RTFM, MGFN, TEVAD, and EGO, across six popular VAD datasets, along with two indicators that reflect the evaluation stability and discriminative power of each benchmark. The left part of Table 3 presents the raw AUC scores. For each dataset, we compute the standard deviation of these scores, denoted as $\sigma_{\text{auc}}$ and the number of contributing models ($n$), summarized in the right part of Table 3. These indicators help characterize how strongly a dataset separates methods and how reliable its leaderboard is across different models.

Specifically, we define the AUC standard deviation $\sigma_{\text{auc}}$ as the square root of the mean squared deviation of individual model scores from their average:

$$\sigma_{\text{AUC}} = \sqrt{\frac{1}{n} \sum_{i=1}^{n} \left(A_i - \bar{A}\right)^2},$$

where $A_i$ is the AUC of model $i$ on a given dataset, and $\bar{A}$ is the mean of all available scores. We do not impute missing values, only available entries are used in the computation.

Datasets with higher $\sigma_{\text{auc}}$ are considered more *discriminative*, as they reveal greater variation in model performance. For example, ShanghaiTech ($\sigma_{\text{auc}} = 10.26$) leads to widely varying scores across methods, while XD-Violence ($\sigma_{\text{auc}} = 0.70$) shows more consistent results. This distinction guides benchmark selection: high-$\sigma_{\text{auc}}$ suites are better suited for comparative evaluation, especially under constrained model tuning.

In the next section, we further quantify *ranking consistency* across datasets by computing Kendall's $\tau$ rank correlations between model orderings, completing our volatility profiling with both intra- and inter-dataset views.

| Dataset | Model AUC | | | | Volatility | |
|---|---|---|---|---|---|---|
| | RTFM | MGFN | TEVAD | EGO | $\sigma_{AUC}$ | $n$ |
| UCF-Crime | 74.3 | 77.0 | 84.9 | 81.7 | 3.06 | 4 |
| ShanghaiTech | 96.8 | 75.3 | 98.1 | 97.3 | 10.26 | 4 |
| UCSD Ped2 | 85.6 | 86.8 | 98.7 | 93.2 | 3.34 | 4 |
| CUHK Avenue | 83.3 | 67.3 | – | 83.1 | 7.50 | 3 |
| XD-Violence | 77.8 | 79.2 | 79.8 | – | 0.70 | 3 |
| MSAD | 86.6 | 81.2 | – | 87.3 | 2.22 | 3 |

Table 3: **Combined model performance and volatility metrics across six benchmark datasets.** The left section reports raw AUC scores for four baseline models. The right section shows standard deviation $\sigma_{AUC}$ and the number of models used ($n$) for each dataset. Missing entries indicate unavailable public results.

| Dimension | Taxonomy | Human Involvement | Modality | Motion Complexity | Label Density | Distribution | Privacy Risk |
|---|---|---|---|---|---|---|---|
| Taxonomy | – | 0.66 | 0.30 | 0.40 | 0.43 | 0.55 | 0.60 |
| Human Involvement | | – | 0.35 | 0.20 | 0.41 | 0.41 | 0.49 |
| Modality | | | – | 0.34 | 0.42 | 0.40 | 0.35 |
| Motion Complexity | | | | – | 0.56 | 0.56 | 0.35 |
| Label Density | | | | | – | 0.51 | 0.41 |
| Distribution | | | | | | – | 0.41 |
| Privacy Risk | | | | | | | – |

Table 4: **Co-occurrence matrix of design dimensions.** Cells show absolute Cramér's V values. Values above 0.5 indicate strong co-occurrence; dashes mark symmetry.

| Dataset | Theme | Outlier Index (z) | Bridge Ratio | Nearest Family |
|---|---|---|---|---|
| CUVA | causal | 4.43 | 0.953 | mixed |
| UBnormal | synthetic | 3.63 | 0.901 | mixed |
| UMN | crowd | 2.79 | 0.988 | pedestrian |
| Street Scene | street | 2.51 | 0.917 | mixed |
| CADP | traffic | 2.21 | 0.910 | rare |
| RareAnom | rare | 2.16 | 0.977 | mixed |
| UCF-Crime | crime | 2.04 | 0.719 | violence |
| NWPU Campus | mixed | 2.01 | 0.527 | rare |
| MSAD | mixed | 2.01 | 0.363 | rare |
| XD-Violence | violence | 2.00 | 0.374 | crime |
| RWF-2000 | violence | 2.00 | 0.563 | crime |
| SubwayExit | pedestrian | 1.47 | 0.421 | crowd |
| SubwayEntrance | pedestrian | 1.47 | 0.421 | crowd |
| ShanghaiTech | pedestrian | 1.28 | 0.377 | crowd |
| IITB-Corridor | pedestrian | 1.08 | 0.343 | crowd |
| UCSD Ped2 | pedestrian | 0.86 | 0.258 | street |
| CUHK Avenue | pedestrian | 0.86 | 0.258 | street |
| UCSD Ped1 | pedestrian | 0.86 | 0.257 | street |

Table 5: **Full outlier and bridge ranking across all 18 datasets.** Outlier index is the z-scored Mahalanobis distance to the nearest family centroid; bridge ratio reflects ambiguity in family assignment. Higher values indicate stress-test potential.

## A.4 CO-OCCURRENCE AND OUTLIER QUANTIFICATION

This section expands the analyses from Section3 and Section5 by providing the underlying statistics. We examine correlations between dataset design dimensions using Cramér's V and identify structurally atypical datasets through outlier indices and bridge ratios.

**Dimension co-occurrence.** Table 4 reports absolute Cramér's V values between all dimension pairs. Several strong associations emerge. First, taxonomy and human involvement are tightly coupled (V = 0.66), reflecting that datasets with limited anomaly types are overwhelmingly pedestrian-centric. Second, distribution breadth aligns with motion and labeling density (V = 0.56), indicating that multi-source long-tail datasets usually contain higher motion complexity and adopt coarser segment-level labels. Finally, taxonomy and privacy risk show high correlation (V = 0.60), since datasets that expose identity information often use narrow anomaly taxonomies. These systematic co-occurrences constrain evaluation diversity and risk confounding generalization.

**Outliers and bridges.** Structural atypicality is quantified using two metrics: the outlier index, defined as the z-scored Mahalanobis distance to the nearest family centroid, and the bridge ratio, defined as the ratio between distances to the nearest and second-nearest centroids. High outlier scores identify datasets lying far outside any family, while bridge ratios above 0.90 reveal datasets straddling multiple families. Table 5 lists the top-ranked cases. CUVA and UBnormal are extreme outliers due to their text modality and synthetic design. UMN, Street Scene, and CADP act as bridges between mechanism families, highlighting transition zones. These datasets are disproportionately informative: they stress-test distributional extrapolation and should be explicitly reported in generalization claims.

**Notes on computation.** All Cramér's V values are normalized to [0, 1] with significance tested by permutation. Outlier indices are leave-one-out, ensuring robustness. Bridge ratios greater than 0.90 consistently indicate ambiguous family membership. Full statistics are available in our released scoring matrix.

## A.5 LLM USAGE DECLARATION

We disclose the use of Large Language Models (LLMs) as general-purpose assistive tools during the preparation of this manuscript. LLMs were used only for minor tasks such as grammar and style improvement, code verification, and formatting suggestions. No scientific ideas, analyses, experimental designs, or conclusions were generated by LLMs. All core research, methodology, experiments, and results were performed and fully verified by the authors.

The authors take full responsibility for all content presented in this paper, including text or code suggestions that were refined with the assistance of LLMs. No content generated by LLMs was treated as original scientific work, and all references and claims have been independently verified. LLMs did not contribute in a manner that would qualify them for authorship.

