# OpenReview forum: "A Dimensional Analysis of Video Anomaly Detection Benchmarks"
_ICLR.cc/2026/Conference — Submitted to ICLR 2026_

### Official Review · Reviewer_Mk9x · 2025-10-23

**Soundness:** 2
**Presentation:** 2
**Contribution:** 2
**Rating:** 2
**Confidence:** 4

**Summary:**

This paper presents a principled framework for analyzing Video Anomaly Detection (VAD) benchmarks by formalizing datasets as seven-dimensional profiles. It introduces structural metrics—Coverage and Balance—and diagnostic indicators to quantify dataset bias and evaluation stability.  The work offers concrete guidelines for more reproducible and socially responsible evaluation, advocating for balanced profiles and standardized reporting of stability metrics.

**Strengths:**

The paper is well organized, and analysis of VLA benchmark is interesting.

**Weaknesses:**

1, after analysis, the authors think a benchmark should hold good coverage and balance, which is a widely acknowledged conclusion. Besides, the author also did not compare the reliability of Coverage and balance indicator with some basic statistics information to access the dataset's coverage and balance.

2, It seems that the proposed indicators can also be applied to the datasets beyond VAM,  nothing special designs for VAM benchmarks.

3,  In conclusion, the authors advices the benchmark with good coverage and balance, but also recommend to include long-tail data for sturctural coverage, is that conflict with the balance?

**Questions:**

see weaknesses

---

### Official Review · Reviewer_5Z2f · 2025-10-26

**Soundness:** 2
**Presentation:** 1
**Contribution:** 1
**Rating:** 2
**Confidence:** 4

**Summary:**

This paper analyzes existing video anomaly detection datasets, attempting to quantitatively characterize datasets along seven different attributes and four proposed axes. Comprehensive statistical analysis is performed to identify correlations between datasets. The authors propose a set of guidelines for evaluating and reporting results in future video anomaly detection works.

**Strengths:**

1. Many different types of statistical analyses are performed, the experiments are comprehensive in this regard.
2. The analysis on benchmark discriminative ability seems like a useful method to help characterize datasets. Coverage and balance are also useful for describing the datasets.

**Weaknesses:**

1. The overall direction and story are quite weak. The idea to better characterize anomaly detection datasets is good, but the overall execution here is confusing and not very actionable. Out of a paper like this, I would expect a concise, profound takeaway for the field moving forward. Instead, we are left with weak guidelines loosely summarized as "show results on more datasets" and "create benchmarks with more types of anomalies and modalities".
2. While the characterization attempts are comprehensive, it is difficult to distill meaningful information from them. Each paragraph essentially reports the metrics shown in the paired figure/graph then provides a one-two sentence plausible interpretation of the metrics. Some of these interpretations are based on an incorrect reading of the results (Lines 314-318, label density/distribution co-occurrence is actually 0.46), making trusting the explanations difficult.
3. The paper is missing a robust description of the different types of video anomaly detection datasets. Correlations defining families are only identified after running the analysis, though some of this can be done without quantified analysis. For example, there is no formal discussion of unsupervised vs. weakly supervised datasets. I suspect readers without comprehensive knowledge of the existing benchmarks would have a very difficult time understanding much of the discussion.
4. Many of the important figures are difficult to distill. For example, in figure one, interpreting a-c takes a lot of back and forth with the text and doesn't convey much insight about the benchmark distributions. This should be a high-impact figure that clearly highlights the important findings of the work.
5. The figure 1 and appendix modality classes are different. RGB vs. RGB + optical flow is a method design choice and is independent of the benchmark itself.

**Questions:**

1. Why have categories such as low human involvement, RGB+Audio+Text, and low motion complexity if there are no datasets that are in those categories.
2. In figure one, what is the meaning of the dendrograms? They are difficult to follow and don't really seem to mean much.

---

### Official Review · Reviewer_JAUd · 2025-10-29

**Soundness:** 3
**Presentation:** 3
**Contribution:** 3
**Rating:** 6
**Confidence:** 4

**Summary:**

This paper presents a systematic dimensional analysis of 18 Video Anomaly Detection (VAD) benchmarks. It argues that existing datasets contain implicit biases (e.g., being human-centric, RGB-only) that distort research focus and limit real-world applicability. The authors introduce a 7-dimension framework and two novel, model-agnostic metrics—Coverage and Balance—to quantify dataset structural properties. The analysis reveals systemic biases across the VAD ecosystem and identifies structural "outliers" (e.g., CUVA, UBnormal) valuable for testing generalization. Critically, the work links these structural properties to evaluation instability, showing that low "Balance" correlates with high cross-model variance and inconsistent leaderboard rankings. It concludes with actionable guidelines for reporting dataset volatility metrics ($\sigma_{auc}$, $\overline{\tau}$) alongside standard performance scores.

**Strengths:**

1.This work introduces a novel 7-dimension framework (e.g., taxonomy, modality, privacy risk) that provides a model-agnostic language for quantifying dataset properties beyond simple size or domain metrics. The resulting metrics, Coverage and Balance, offer a new, quantitative method for dataset selection and for integrating ethical considerations (like the "Privacy Risk" dimension) directly into the analysis.

2.The paper provides clear, quantitative evidence of systemic biases in VAD, such as the over-reliance on RGB-only data. The co-occurrence analysis (Fig. 3, Table 4) is highly insightful, uncovering potential model "shortcuts," like the strong correlation between "taxonomy level" and "human involvement". The identification of "outlier" and "bridge" datasets (Fig. 5) provides an immediate, actionable guide for stress-testing model generalization.

3.The paper links low structural "Balance" to high cross-model score variance (Fig. 4b), providing a structural reason for unstable leaderboards. Furthermore, the analysis of "discriminative power" ($\sigma_{auc}$, Fig. 6) and rank stability (Kendall's $\tau$, Fig. 7) gives researchers practical tools to identify which benchmarks are most effective for separating models (e.g., ShanghaiTech) and which produce unstable rankings (e.g., CUHK Avenue).

**Weaknesses:**

1.The 7-dimension framework relies on ordinal scales with qualitative boundaries (e.g., "medium" vs "high" motion complexity) that are subject to annotator interpretation. The paper states two annotators were used but critically fails to report any inter-annotator agreement (IAA) metrics (e.g., Krippendorff's Alpha). This omission makes it difficult to assess the reliability of the foundational data in Table 2.

2.The core empirical validation in Sec 5 is based on a small sample of only four baseline models (RTFM, MGFN, TEVAD, EGO). This small set may not capture the diversity of modern architectures. This is compounded by using only six datasets with several missing data points, which reduces the statistical power and confidence in the correlation analyses presented in Fig. 4 and Fig. 7.

3.The "Coverage" and "Balance" indicators are presented as heuristic constructions. The paper does not theoretically justify the choice of these specific metrics (e.g., Shannon entropy for "Balance") over other established measures of diversity or dispersion (e.g., Gini impurity, statistical variance, etc.). Furthermore, there is no sensitivity analysis provided. It is unclear how robust these metrics are to changes in the framework, such as the number of dimensions used or the discrete number of levels (e.g., 4 vs. 5) chosen for each dimension.

4.The paper's excellent proposal that future work report the $(\sigma_{auc}, \overline{\tau}, n)$ tuple lacks a key component for standardization. The metrics $\sigma_{auc}$ and $\overline{\tau}$ are highly dependent on the set of baseline models ($n$) used in the calculation. Because the paper does not propose a "standard basket" of 3-5 representative baselines, these metrics will not be comparable across different papers, as each paper may choose a different $n$. This makes the recommendation difficult to adopt in a standardized way.

**Questions:**

1. Could the authors please address the reliability of their scoring? Specifically, (a) what was the inter-annotator agreement (IAA) score (e.g., Krippendorff's Alpha) for the dataset-by-dimension matrix (Table 2)? and (b) how were qualitative boundaries like "medium" vs "high" motion complexity consistently adjudicated?

2. Given the small and incomplete sample (n=4 models, n=6 datasets, with missing values), how confident are the authors in the generality of the correlations found in Fig. 4 and Fig. 7? How might these results change if more diverse, modern architectures were included?

3. Regarding the "Balance" metric, could the authors comment on why normalized Shannon entropy was chosen over other potential measures of statistical dispersion (like Gini impurity or variance)? Was any sensitivity analysis performed on the robustness of these metrics?

4. For the proposed $(\sigma_{auc}, \overline{\tau}, n)$ tuple to be a comparable metric across papers, the set of baseline models ($n$) would need to be standardized. Can the authors explain and analyze what would constitute a minimal, representative 'basket' of models to serve this purpose? This analysis is not specified in the manuscript but is crucial for the proposal's practical adoption.

---

### Official Review · Reviewer_c5Um · 2025-10-31

**Soundness:** 3
**Presentation:** 2
**Contribution:** 2
**Rating:** 4
**Confidence:** 3

**Summary:**

This research, Proposes a 7-dimension profiling framework for VAD datasets (taxonomy, human involvement, modality, motion complexity, label density, distribution, privacy). Introduces Coverage (breadth of levels used) and Balance (normalized entropy of level usage) as dataset-level indicators. Evaluates 18 datasets, visualizing structure (matrix + t-SNE) and identifying dataset “families,” outliers, and bridges. Finds strong RGB/pedestrian bias, limited multimodality, sparse temporal labels, and single-location concentration. Links dataset structure to evaluation stability via AUC variance and mean Kendall’s ; recommends reporting these alongside headline AUC/AP.

**Strengths:**

1. Turns qualitative dataset critique into auditable, reproducible profiles.
2. Clear reporting checklist suggestions, highlights outlier/bridge sets for testing generalization.
3. Matrix, clusters, and metadata make comparisons intuitive, and Framework applies regardless of the underlying VAD approach.

**Weaknesses:**

1. Mapping datasets to ordinal levels relies on human judgment, and limited detail on inter-annotator agreement.
2. Scope limits: Volatility analysis uses a small set of baselines/datasets—generalization of correlations may be brittle.
3. Granularity gaps: Some dimensions (e.g., “distribution,” “privacy”) can be coarse and context-dependent.

**Questions:**

1. Can you release the scoring rubric with inter-annotator reliability (e.g., κ) and versioned logs?
2. How do Coverage/Balance and volatility metrics behave with more (and newer) baselines, including multimodal/foundation models?
3. Could some dimensions be automated (e.g., modality checks, motion statistics) to reduce subjectivity?

---

### Meta-Review · Area_Chair_UaE8 · 2025-12-06

**Summary:**

This paper has received mixed evaluations, with scores of 4, 6, 2, and 2. Although one reviewer is positive, the other three lean toward rejection. Across the reviews, several major concerns are consistently raised: the reliability of the manually annotated dataset, the small size of some experimental datasets, the lack of theoretical analysis about the metrics, and issues with the overall readability and clarity of the paper. Furthermore, there is no rebuttal from the authors to address these issues. Considering these unresolved concerns, the AC decided to reject this paper.

**Reviewer Concerns:**

There is no rebuttal from the authors. The reviewers raised concerns about the reliability of the manually annotated dataset, the small size of some experimental datasets, the lack of theoretical analysis about the metrics, and issues with the overall readability and clarity of the paper.

**Reviewer Scores:**

Since there is no rebuttal from the authors, it is not possible to reasonably predict whether any of the reviewers would have revised their scores.

---

### Decision · Program_Chairs · 2026-01-26

Reject